# Patient and caregiver characteristics associated with differential use of primary care for children and young people in the UK: a scoping review

Kevin Herbert ![ORCID],[1] Lauren Herlitz ![ORCID],[2] Jenny Woodman ![ORCID],[3] Claire Powell ![ORCID],[2] Stephen Morris ![ORCID] [1]

¹Cambridge Research Methods Hub, Department of Public Health and Primary Care, Cambridge University, Cambridge, UK
²Population, Policy and Practice Department, UCL GOS Institute of Child Health, London, UK
³Institute of Education, UCL Social Research Institute, London, UK

**Correspondence to**
Dr Kevin Herbert;
kch28@medschl.cam.ac.uk

## ABSTRACT

**Objective** To systematically map evidence to answer the research question: *What is the relationship between the characteristics of children and young people (CYP) or their caregivers and primary care service use in the UK, taking into account underlying healthcare needs?*
**Design** Scoping review.
**Setting** Primary care.
**Eligibility criteria** English-language quantitative or mixed-methods studies published between 2012 and 2022.
**Data sources** Medline, Embase, Scopus and Web of Science Social Sciences Citation Index, and grey literature.
**Results** 22 eligible studies were identified, covering general practice (n=14), dental health (n=4), child mental health (MN) services (n=3) and immunisation (n=1). Only eight studies (36%) controlled for variables associated with healthcare need (eg, age, birth weight and long-term conditions). In these, evidence of horizontal inequity in primary care use was reported for CYP living in deprived areas in England, with and without complex needs. Horizontal inequity was also identified in primary care MN referrals for CYP in England identifying as mixed-race, Asian or black ethnicity, compared with their white British peers. No evidence of horizontal inequity was observed, however, in primary care use for CYP in England exposed to parental depression, or for CYP children from low-income households in Scotland. Increasing CYP's age was associated with decreasing primary care use across included studies. No studies were found regarding CYP from Gypsy or Traveller communities, children in care, or those with disabilities or special educational needs.
**Conclusions** There is evidence that socioeconomic factors impact on CYP's primary care use, in particular age, ethnicity and deprivation. However, better quality evidence is required to evaluate horizontal inequity in use and address knowledge gaps regarding primary care use for vulnerable CYP populations and the impact of policy and practice related 'supply side' of primary care.

## INTRODUCTION

In recent years, health inequality (avoidable differences in health outcomes between groups) has been growing among the UK population.[1,2] For children and young people

---

**STRENGTHS AND LIMITATIONS OF THIS STUDY**

⇒ Through the detailed data extraction of included studies, this review comprehensively documents the study populations, healthcare settings and CYP/caregiver characteristics considered to explore the use of primary care by children and young people (CYP) in the UK.
⇒ By restricting eligible publications to the last decade, the findings reported are relevant to current healthcare systems and social context.
⇒ Through the use of systematic methodology—including publication quality appraisal—this review was designed and conducted to ensure robustness and reproducibility.
⇒ As a scoping review, this review and its findings should not, however, be treated as exhaustive.

---

(CYP) in England, those from Pakistani, black African and black Caribbean ethnic groups experience higher rates of infant mortality than other ethnicities.[1] Vulnerable CYP (eg, those with a learning disability or autism) have worse health and well-being outcomes, service experiences, poor outcomes associated with chronic conditions and greater premature mortality risk.[3–6] In recognition, the National Health System (NHS) England long-term plan highlights the role of primary care in reducing health inequalities.[7]

Horizontal inequity is defined as the unequal treatment of people with equal need.[8] Despite evidence of horizontal inequity across the UK adult population,[9,10] inequities among CYP are less well understood, particularly in primary care. This evidence gap is of particular concern given its potential to inhibit the effective development of policies to ensure that healthcare provision effectively meets CYP's level of healthcare need.

Analysis of routinely collected national data indicates that infants and preschool children have the highest general practice visit rates of

any age group,[11 12] while in the year 2021–2022, nearly half of children aged 0–17 years in England receive NHS dental care annually.[13] Beyond dental service access by patient age, however, there are no government-provided national CYP statistics on the use of primary care use, or how their use varies for different sub-populations relative to healthcare need.

In England, policies have been introduced over the last 30 years to specifically tackle deprivation-based inequity in general practice.[14] Despite some evidence of improvements, their effectiveness may have been impaired by inadequate compensation for additional deprivation-related healthcare needs via the core general practice funding formula.[14]

To effectively address equity within primary care and inform policy-making relating to the supply of care (eg, service quality, quantity and distribution) the best available evidence is needed for both vulnerable and marginalised CYP groups and the CYP population as a whole. In this article, we examine existing evidence on CYP primary care use and ask: what is the relationship between the characteristics of CYP or their caregivers and primary care service use in the UK, taking into account underlying healthcare needs?

## METHOD

The design of the review was informed by the Preferred Reporting Items for Systematic Reviews and Meta-Analysis extension for scoping reviews (PRISMA-ScR) guidance for rapid evidence reviews, developed by Tricco *et al.*[15] The protocol was registered in the Open Science Framework (Centre for Open Science, https://osf.io/mfc3z) and followed the PRISMA-ScR guide for the review design and reporting of methods and findings, and associated checklist.[16] Due to the rapid nature of the review (10 weeks), the questions and search strategy were targeted to identify relevant articles that could be analysed within the review timeframe.

### Inclusion/exclusion criteria

A study was included if it:
- ▶ Focused on the characteristics of CYP (aged 0–25 years) or families of children (aged 0–18 years);
- ▶ Was conducted on data for UK primary healthcare settings (general practice, community pharmacy, NHS-provided dentistry);
- ▶ Outcomes were reported in terms of primary care use measures (eg, primary care contacts, NHS dentist registration);
- ▶ Used quantitative or mixed-methods empirical methods;
- ▶ Was published between 1 January 2012 and 5 July 2022.
  Studies were excluded if they:
- ▶ Focused on school-based health services (eg, school nursing).

### Search strategy

We used free-text and controlled terms (online supplemental appendix A) to search four electronic databases Medline, Embase, Scopus and Web of Science (Social Sciences Citation Index). To identify publications not indexed in these main databases, an additional 100 publications identified from a grey literature search using Google Scholar, were screened.[17 18]

### Document selection

The search results were imported into Rayyan software (https://www.rayyan.ai/) for de-duplication, before being divided equally between two reviewers and screened independently by title and abstract. 10% of excluded articles for each reviewer were cross-checked by the other, to confirm agreement. As all excluded articles were agreed between reviewers at this stage, screening proceeded to a full-text review by a single reviewer, with any articles for query referred to a second reviewer.

### Data extraction and quality assessment

From included studies, two reviewers extracted: study sample/population; healthcare setting; area of healthcare; study design; methodology; variables controlling for healthcare need (where applicable) and primary care contacts. We also extracted data for outpatient attendance, specialist referrals and hospital-based service use (eg, emergency presentations; hospital admissions) to explore potential associations between CYP primary care use and other health services. Study quality was assessed using the Mixed Methods Appraisal Tool (MMAT).[19 20] No study was excluded based on quality, but we report study quality.

### Data synthesis

As a scoping review, assessment of study heterogeneity and meta-analysis of study results were not performed. Study findings on CYP primary care service use and any associations with patient or caregiver characteristics were narratively synthesised.

### Patient and public involvement

As a scoping review of published literature, no patients were involved in the design, conduct or analysis of this research.

## RESULTS

We found 2301 unique title/abstracts, of which 22 publications (reporting 21 studies) met the inclusion criteria (figure 1).

### Study characteristics

#### Study origin, healthcare setting and area of health

Most studies related to English primary care services (n=17); the rest studied primary care in Northern Ireland (n=2), Wales (n=1), Ireland and Scotland (n=1) and the UK (n=1) (see online supplemental appendix B). Most focused on utilisation of general practice services (n=14),

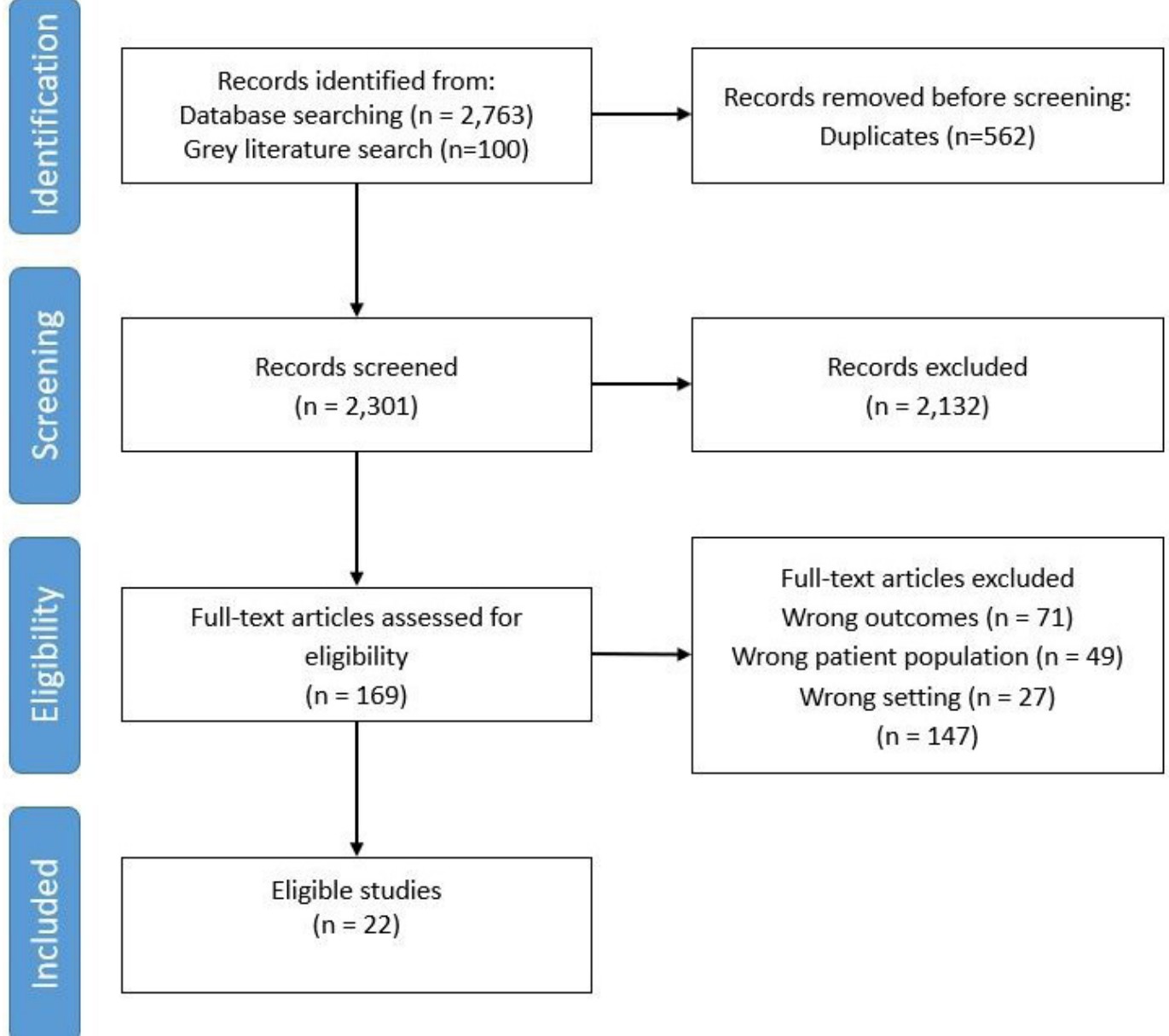

**Figure 1** Preferred Reporting Items for Systematic Reviews and Meta-Analyses flow diagram: associations between children and young people or caregiver characteristics and primary care use.

with the remaining, dental services or oral health (n=4), child mental health (MH) services (n=3) and the uptake of immunisations (n=1).

### Study designs and samples

Most of the eligible publications reported on cohort studies (n=17), with the remaining cross-sectional studies (n=5). The most common age ranges of CYP studied were 0–15 years (n=3), 0–18 years (n=3), 0–1 years (n=2) and 0–14 years (n=2) (see figure 2 and online supplemental appendix B).

### Characteristics studied

Eligible studies predominately investigated deprivation or social-economic differences (n=16), sex or gender

(n=13), ethnic group (n=11) or age (n=8) for potential associations with levels of primary care service use. Five of the included studies compared primary care use for CYP with a defined health condition: congenital abnormalities; CYP MN; Down's syndrome and eczema (online supplemental appendix B).

15 of the included studies contained one or more healthcare need variables in their analysis of CYP primary care use (online supplemental appendix B). Eight studies controlled for healthcare need in a general population sample, with CYP age (n=8), gestational age (n=3) and birth weight (n=2) the most often controlled for in regression analyses. Seven studies compared use between subsets of the primary CYP sample, the most commonly

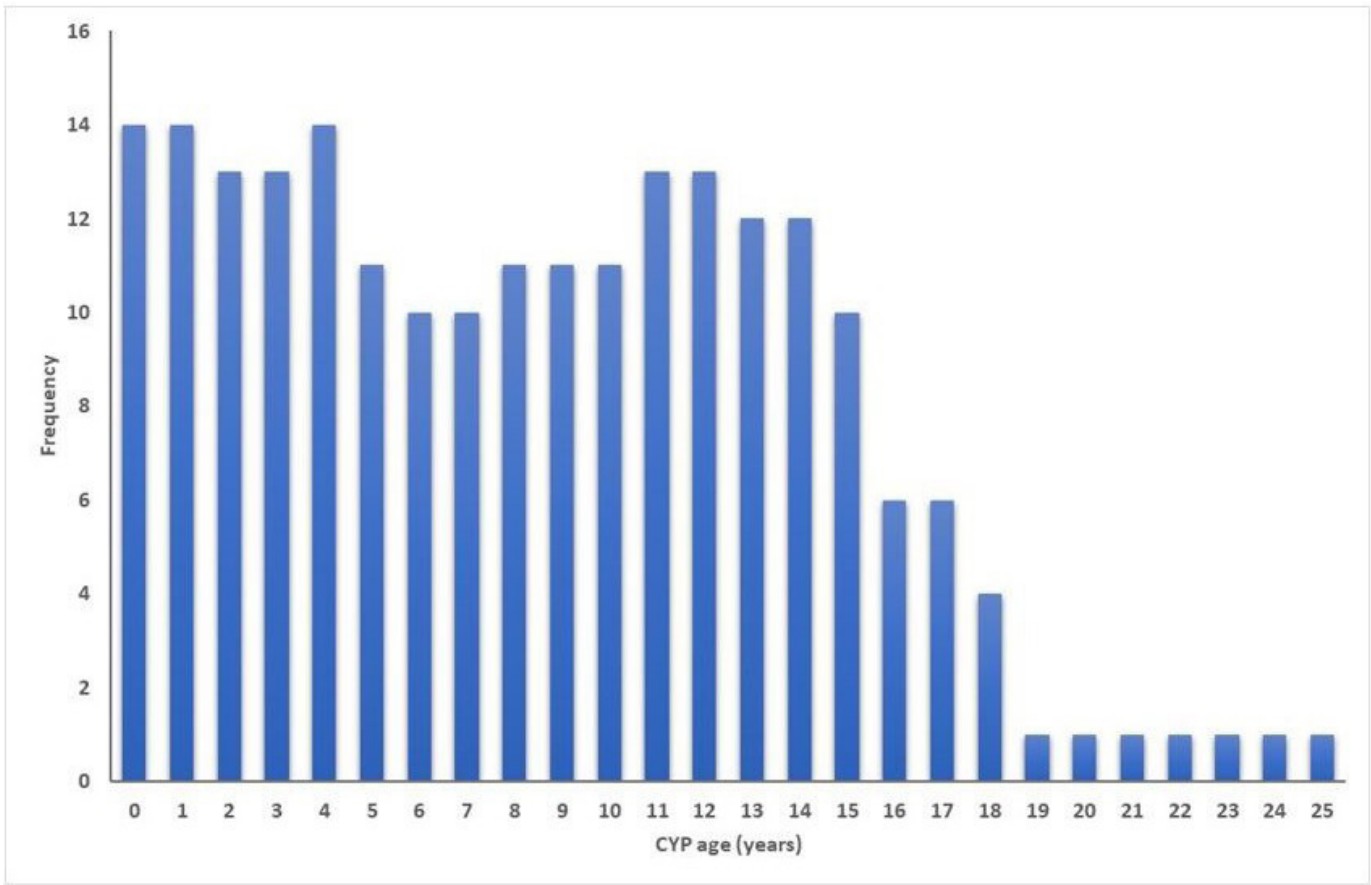

**Figure 2** Frequency distribution of CYP ages included in eligible studies. CYP, children and young people.

investigated subsets were CYP age (n=7), CYP with comorbidities (n=2) or CYP with defined conditions (n=2). None of the included studies accounted for variation in supply (eg, CYP distance from practices, general practitioner (GP) appointment availability, GP to patient ratio) in the analysis of CYP use of primary care.

### Outcomes
Table 1 summarises healthcare utilisation outcomes reported by included studies. The services most investigated were primary care attendance and/or GP consultations (n=13), outpatient attendance (n=7) and specialist referrals (n=5). Use of non-primary care services was recorded in nine of the eligible studies (see table 1), to study their use in addition to primary care or to explore possible relationships between primary care presentations and the need for hospital-based care.

The relationship between CYP characteristics and use of publicly funded dental health services was investigated in four of the included studies. Outcomes reported for these studies focused on the registration or rates of attendance with NHS (England) or General Dental Service (Northern Ireland) dental healthcare services, dental or oral health by examination, or publicly funded fee for service reimbursement for dental care provided (see table 1).

### Study quality
All studies bar one met four or five of the MMAT quality criteria (see online supplemental appendix B). 15 of the 22 eligible publications included a statistical analysis and/or a study sample which accounted for healthcare need. Of these, eight high-quality publications controlled for healthcare need in their analysis. The remaining seven publications were assigned high quality as their study designs were appropriate for their respective research questions, but we noted that indicators of healthcare need as criteria were only used for deriving study samples (for full quality criteria, see online supplemental appendix C).

### CYP or caregiver characteristics affecting primary care use
The next sections report the CYP or caregiver factors associated with UK primary care use. Although some characteristics were examined in multiple studies (eg, CYP age) or with large samples (eg, parental MN), overall, the literature was fragmented with one or two studies reporting findings for some characteristics. Findings related to general practice are reported first in each section, followed by use of dental services. We have highlighted findings from high-quality studies which have controlled for healthcare need variables.

**Table 1** Outcomes reported in included studies

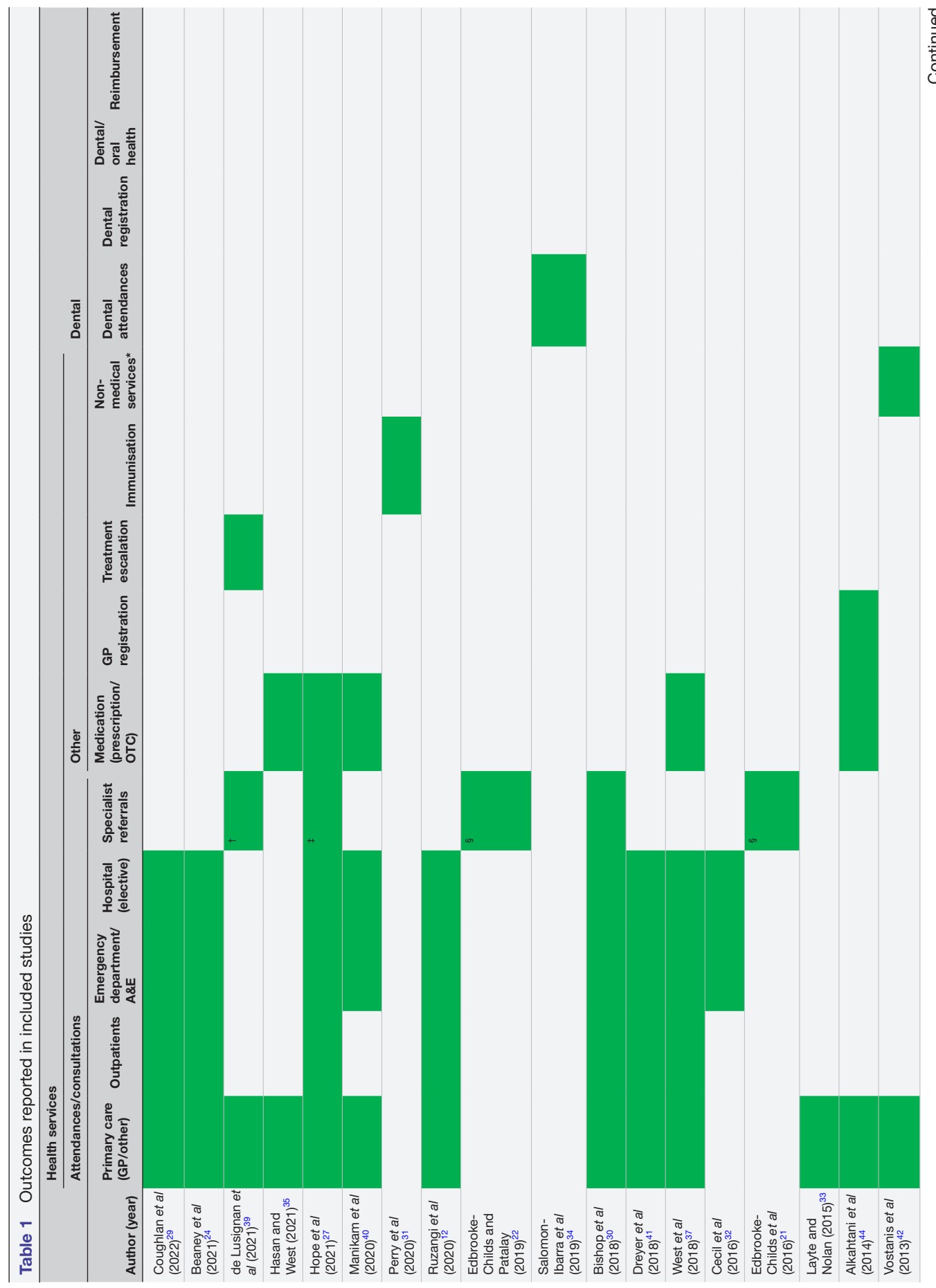

| Author (year) | Health services | | | | Other | | | | | | Dental | | | Reimbursement |
| | Attendances/consultations | | | | | | | | | | | | | |
| | Primary care (GP/other) | Outpatients | Emergency department/A&E | Hospital (elective) | Specialist referrals | Medication (prescription/OTC) | GP registration | Treatment escalation | Immunisation | Non-medical services* | Dental attendances | Dental registration | Dental/oral health | |
|---|---|---|---|---|---|---|---|---|---|---|---|---|---|---|
| Coughlan et al (2022)[29] | | | | ✓ | | | | | | | | | | |
| Beaney et al (2021)[24] | | | | ✓ | | | | | | | | | | |
| de Lusignan et al (2021)[39] | ✓ | | | | ✓† | | | ✓ | | | | | | |
| Hasan and West (2021)[35] | ✓ | | | | | ✓ | | | | | | | | |
| Hope et al (2021)[27] | ✓ | | | | ✓‡ | ✓ | | | | | | | | |
| Manikam et al (2020)[40] | ✓ | | ✓ | | ✓ | ✓ | | | | | | | | |
| Perry et al (2020)[31] | | | | | | | | | ✓ | | | | | |
| Ruzangi et al (2020)[12] | | | | ✓ | ✓§ | | | | | | | | | |
| Edbrooke-Childs and Patalay (2019)[22] | | | | | ✓ | ✓ | | | | | | | | |
| Salomon-Ibarra et al (2019)[34] | | | | | | | | | | | ✓ | | | |
| Bishop et al (2018)[30] | ✓ | | | | ✓ | ✓ | | | | | | | | |
| Dreyer et al (2018)[41] | | | | ✓ | ✓ | | | | | | | | | |
| West et al (2018)[37] | | | | ✓ | | | ✓ | | | | | | | |
| Cecil et al (2016)[32] | | ✓ | ✓ | ✓ | ✓ | | | | | | | | | |
| Edbrooke-Childs et al (2016)[21] | | | | | ✓§ | ✓ | | | | | | | | |
| Layte and Nolan (2015)[33] | ✓ | | | | | | | | | | | | | |
| Alkahtani et al (2014)[44] | ✓ | | | | | ✓ | | | | | | | | |
| Vostanis et al (2013)[42] | ✓ | | | | | | | | | ✓ | | | | |

Continued

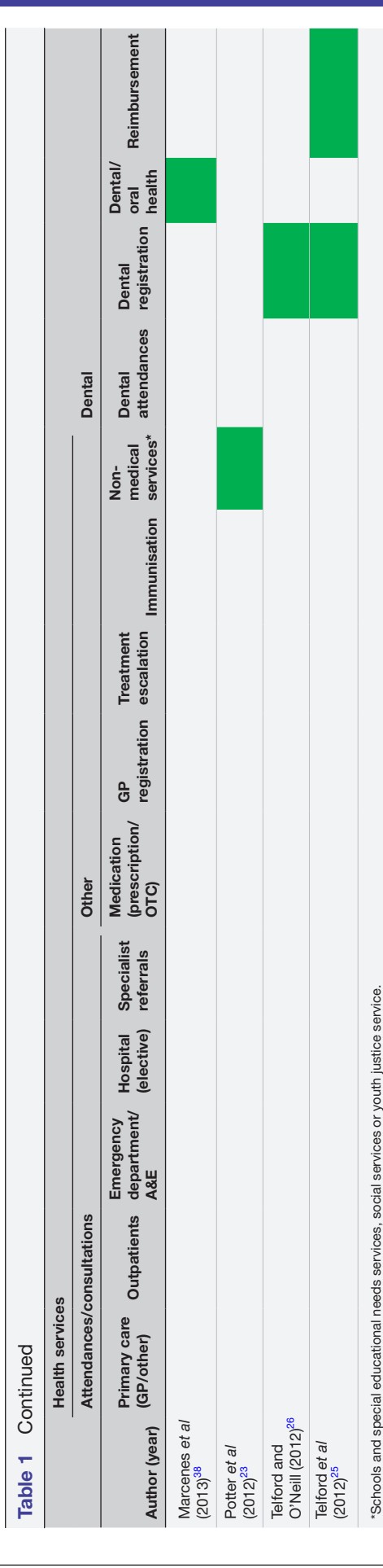

**Table 1** Continued

*Schools and special educational needs services, social services or youth justice service.
†Dermatologist, GP with a specialty interest in dermatology or dermatology specialist nurse.
‡ Not specified.
§ Mental health services referrals.
A&E, Accident and Emergency; GP, general practice; OTC, over the counter.

## Sex or gender of CYP

Some evidence was apparent for gender-based differences in the use of primary care, and the routes by which CYP are referred to specialist services. Two high-quality studies that controlled, respectively, for age, and age and type of MN problem found that males were more likely to be referred to child and adolescent mental health services (CAMHS) via education services than via primary care, compared with their female counterparts.[21 22] Female CYP with a parental history of recurrent depression were more likely to use primary care for the treatment of their own anxiety disorder than their male counterparts.[23] In contrast, in a study that characterised different patterns of CYP healthcare use in Northwest London, males were more likely to appear in the high service use cluster compared with females, after adjusting for age, ethnicity, deprivation and long-term conditions.[24]

Where dental services were studied, females were found to be more likely to use orthodontic services than males.[25] Access to dental services also became differentiated with age with females being more likely to be registered with an NHS dentist than males by the age of 15–16 years, with no difference at 11–12 years of age.[26]

## Age of CYP

Consistent across included studies, increasing age of the child was associated with a fall in the use of primary care services, with older teenagers less likely to use primary care.[12 24 26–28]

## Deprivation or socioeconomic classification

CYP from areas of higher deprivation in England were reported to have had lower levels of GP consultations, considering both CYP in the general population or CYP with congenital abnormalities.[29 30] This was despite the overall need for healthcare being higher, reflected in a shift from the routine use of scheduled care services (eg, GP consultations and outpatient attendance) to those that deal with unscheduled presentations (eg, emergency department attendance) or hospital admissions. Exposure to deprivation was also observed to affect the uptake and timeliness of child vaccinations, with lower rates of uptake and later vaccinations for children from areas of highest deprivation.[31] Of note, family GP practices with highest reported accessibility according to the GP Patient Survey were more likely to be located in more affluent areas, less likely to be in urban areas, and also had a lower proportion of their registered patient list being children.[32]

In contrast to the above, the study of different patterns of healthcare use in Northwest London found higher levels of deprivation in clusters of CYP that had higher health service utilisation, including primary care.[24] This finding indicates that a minority of CYP from the most deprived areas may be frequent attenders of primary care, but in general, more CYP from deprived areas would be expected to have higher rates of primary care service use due to higher levels of healthcare need. It should also be considered that the Beaney *et al*[24] study may not be

nationally representative of the UK CYP population, due to the study population being drawn from an area with higher ethnic diversity and levels of deprivation, relative to national averages.[24]

One high-quality study that accounted for CYP age, birth weight and parental assessment of CYP health among other factors explored family income as a possible factor in inequality of GP service utilisation for samples of children from Scotland and the Republic of Ireland.[33] We report only on the UK-related findings. The authors identified little or no overall income-related inequity in GP care for Scottish children in the overall analysis. The probability of GP visits was higher in younger children (2-year-olds) from less deprived backgrounds, although this was almost totally negated after adjustment for health and non-need determinants.

In dental care, children in the lowest socioeconomic status group were more likely to require reparative dental treatments, and dental treatment overall, but were less likely to have consumed orthodontic services.[26 34]

For children with congenital abnormalities, higher maternal education was associated with lower levels of primary care consultations and hospital service use—potentially a consequence of less educated parents requiring additional help from medical professionals to manage CYP complex needs.[30] In contrast, lower educational attainment of the household reference person was associated with CYP having less time registered with an NHS dentist,[26] and where the reference person had no qualifications, more than twice was spent on endodontics for the CYP than where a degree or greater was obtained.[25]

### Ethnic background

Associations were identified between ethnic background and primary care use, which were moderated by intersectionality. Three high-quality studies, including one that controlled for age, birth weight and gestational age,[35] reported that children from black, Pakistani, Asian or Asian British ethnicity were observed to have GP consultation or prescription rates at higher levels than the majority population, while outpatient attendance and other services were relatively underused.[24 29 35]

Babies born at term with low birth weights are recognised as having increased morbidity and mortality through infancy to adulthood.[36] Some ethnic minorities (Indian, Bangladeshi, Pakistani, black Caribbean or other) are observed to be over-represented in terms of babies delivered at low birth weights.[1] However, one high-quality study that accounted for CYP gestational age found that children of Pakistani ethnicity had greater primary care use than white British children irrespective of birth weight.[37]

Two high-quality studies that controlled respectively for age and type of MN problem found that children of black, Asian, other (non-white), white other or of mixed race were more likely to be referred to child MN services through routes other than primary care (eg, education services, social services, youth justice), relative to their white British counterparts.[21 22] Children of Asian, black, white other or other ethnicities were also less likely to have their case closed through non-attendance—a finding thought to be related to the higher rate of compulsory child MN referrals.[21 22]

Dental services reflected poorer outcomes in children of ethnic minorities, with those of white European, Bangladeshi or Pakistani ethnicity more likely to have poor dental health (tooth decay, teeth missing due to caries, fillings score or greater numbers of untreated carious teeth), relative to children identifying as white British.[38]

### CYP with existing conditions

Differences in primary care service use were observed for children with existing conditions. One high-quality study that controlled for multimorbidity found that CYP of Pakistani or 'other' ethnic origin with congenital abnormalities were found to have higher primary care use compared with their white British peers.[30] A greater chance of eczema treatment escalation was also seen for CYP of Asian, black African/Caribbean, mixed or other ethnicities, relative to those of white ethnicity.[39] It is possible that these variations may be reflective of condition-specific differences for different population subgroups; the performance of treatment pathways; GP–patient relationships; parental attitudes or their presentation thresholds for primary care attendance. For children with Down's syndrome (DS), higher rates of primary care use (GP consultations, prescriptions) due to respiratory tract infections (RTIs) were observed, relative to a matched control group without DS.[40] The authors noted that this may at least in part be due to increased RTI susceptibility and severity in children with DS, arising from vulnerabilities caused by differences in immunology, and airway morphology and function.

### Parental MH

Two high-quality studies, one of which accounted for CYP's age and presence of chronic conditions, found that CYP exposure to maternal mental illness was found to be more prevalent with increased deprivation and associated with increased CYP primary healthcare use.[27 41] Maternal mental illness exposed CYP had higher rates of primary care service contacts, outpatient attendance and specialist referrals. They also received more drug prescriptions for mental and behavioural disorders, physical, acute or chronic diseases. Across all CYP who accessed primary care services, use was greater for those with mothers who had poor MN than those without, which may indicate differences in health-seeking behaviour or a higher level of healthcare need.[27 41]

In contrast, Potter et al[23] did not find any effect of having parents with current MN conditions on the use of primary care by CYP.[23] It should be noted, however, that this study was based on a relatively small sample size (n=333), compared with those used in Hope et al[27] and Dreyer et al[41] (n=489 255 and n=25 252, respectively).

### Migrant status

Eligible studies for primary care access and use by children with migrant backgrounds were limited. One high-quality study that controlled for CYP age reported that children of Indian ethnicity with a migrant background were less likely to self-refer for MN issues, although no differences in overall use of MN services were observed.[42 43]

### Refugee status

Only one study (of lower quality) was identified on primary care use by a sample of children with refugee status of varied ethnicities (Afghanistan, Ethiopia, Gambia, Iran, Iraq, East Africa, Kenya, Nigeria, Pakistan, Somalia, Sudan, Tunisia, Vietnam, Zambia, Zimbabwe). No difference was found for medicines consumption relative to the control group studied,[44] although evidence did indicate that over the counter, rather than prescribed medicines were more likely to be the primary source for refugee children.

## DISCUSSION

While the review identified multiple studies of CYP primary care use, relatively few took into account indicators of CYP's underlying healthcare needs. Studies consistently found lower rates of CYP primary care use with increasing CYP age. There was evidence that CYP living in deprived areas in England used primary care less than their counterparts living in wealthier areas, a finding identified by both studies that accounted for indicators of underlying healthcare needs and those that compared subsets of CYP. There was some evidence to suggest that CYP in deprived areas were more likely to use acute care services. Recent research has identified key principles to reduce the likelihood of health inequalities resulting from general practice, including accounting for differences within patient groups.[45]

Included studies also suggested horizontal inequity in referrals to MN services. Primary care services were less likely to refer male or non-white British CYP to CAMHS, potentially as a result of differences in help-seeking preferences, patient engagement or 'GP gatekeeping',[46] highlighting an issue for which service and/or patient-focused solutions are being sought.[47–49] Where NHS dental services were studied, a need for greater oral care education and outreach was indicated by the poor dental health and service access for CYP of non-white British ethnicity, and those from low socioeconomic status and poorly educated households.[26 34 38]

Two-thirds of the studies (n=14) did not control for underlying healthcare need in their analyses, and as a consequence it is difficult to understand whether differences in primary healthcare use reflect service inequalities or variation that is appropriately based on differences in need. Where studies did account for health indicators, age, gestational age and birth weight were the most commonly controlled for variables. Morbidity or CYP health status was largely absent from analyses. With regard to other factors that may influence CYP primary care use, little reference was made to supply-side issues in general practice (eg, appointment availability, practice location, GP-to-patient ratio), outside of a single study focused on accessibility via responses to the GP Patient Survey. While the growing pressures on general practice are well reported,[50] how these affect the CYP population and specifically how they may drive inequities in service use remains unclear.

Key gaps in the knowledge landscape were also identified. It should be noted that all included studies focused on CYP registered with primary care providers, with no studies found for unregistered patients, or service use in other care settings (eg, walk-in centres), pointing towards a paucity of data collection in these areas. Little quantitative evidence was found regarding CYP primary care use in refugee or migrant populations, while no studies investigated primary care use for CYP from Gypsy or Traveller communities, care-experienced CYP, or for CYP with learning disabilities, autism or special educational needs.

This review was strengthened through clearly defined scoping review protocols, thus optimising search, screening and data extraction processes and ensuring robustness and reproducibility. Through the detailed extraction of data from included studies, this review comprehensively documents the study populations, healthcare settings and CYP/caregiver characteristics considered to explore the use of primary care by CYP in the UK. By confining eligible publications to those from within the last decade, this study also focuses on evidence relevant to current social dynamics and clinical practice.

To inform the wider research project on CYP's access and use of primary care, we took a pragmatic approach to the timeframe for the literature review, conducting the review rapidly while retaining elements required for a robust review. It is understood that there is debate regarding the value of Google Scholar in systematic reviewing;[17] however, it was felt important to include a search of grey literature, due to the focus of the study—aspects of which (eg, horizontal inequity) are of particular interest to organisations who produce research reports which would not fall within the scope of the key literature databases. Limiting the main search to key literature databases maximised the coverage of potentially relevant publications in the time available and allowed for detailed data extraction and analysis from the publications identified. Consequently, the findings should not be treated as exhaustive.

## CONCLUSION

Recent evidence suggests that CYP age, deprivation, ethnicity and gender influence CYP primary care use, including NHS dental services. Studies that have accounted for indicators of healthcare need suggest that CYP living in deprived areas in England use primary care services less than CYP in wealthier areas with equal levels of need and have greater use of acute care. Despite these

findings, our assessment is that better quality evidence is required to adequately evaluate horizontal inequity in CYP primary care use and address knowledge gaps regarding primary care use for vulnerable CYP populations, as well as the impact of 'supply side' policy and practice for the delivery of high-quality primary care services to all CYP.

**Contributors** We attest that all authors contributed significantly to the creation of this manuscript, each having fulfilled the criteria as established by the ICMJE. Specifically: KH (Guarantor): Substantial contribution to the design of the work, the acquisition, analysis and interpretation of data for the paper and the drafting and revision of the submitted paper. They agree to be accountable for all aspects of the work, and in ensuring that questions related to the accuracy or integrity of any part of the work are appropriately investigated and resolved. LH: Substantial contribution to the design of the research, the analysis and interpretation of data for the submitted paper and the critical review and revision of the paper for important intellectual content. JW: Substantial contribution to the conception and design of the research, the critical review and revision of the paper for important intellectual content and final approval of the version to be published. They agree to be accountable for all aspects of the work, and in ensuring that questions related to the accuracy or integrity of any part of the work are appropriately investigated and resolved. CP: Substantial contribution to the design of the research, the analysis and interpretation of data for the submitted paper and the critical review and revision of the paper for important intellectual content. SM: Substantial contribution to the conception and design of the research, the analysis and interpretation of data for the submitted paper, the critical review and revision of the work for important intellectual content and final approval of the version to be published. They agree to be accountable for all aspects of the work, and in ensuring that questions related to the accuracy or integrity of any part of the work are appropriately investigated and resolved. We as authors also confirm that the manuscript has been read and approved by all named authors and that the order of authors listed in the manuscript has been approved by all named authors.

**Funding** This study is funded by the National Institute for Health and Social Care Research (NIHR) through the Children and Families Policy Research Unit (PR-PRU-1217-21301). The views expressed are those of the authors and not necessarily those of the NIHR or the Department of Health and Social Care.

**Competing interests** None declared.

**Patient and public involvement** Patients and/or the public were not involved in the design, or conduct, or reporting, or dissemination plans of this research.

**Patient consent for publication** Not applicable.

**Ethics approval** As a scoping review, this research did not involve participants and thus ethics approval was not required.

**Provenance and peer review** Not commissioned; externally peer reviewed.

**Data availability statement** All data relevant to the study are included in the article or uploaded as supplementary information.

**ORCID iDs**
Kevin Herbert http://orcid.org/0009-0008-4354-7811
Lauren Herlitz http://orcid.org/0000-0003-2497-9041
Jenny Woodman http://orcid.org/0000-0002-9403-4177
Claire Powell http://orcid.org/0000-0002-6581-0165
Stephen Morris http://orcid.org/0000-0002-5828-3563

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
