## [Reviewer comments · BMJ Open]

ARTICLE DETAILS

TITLE (PROVISIONAL)	Patient and caregiver characteristics associated with differential use of primary care for children and young people in the UK: a scoping review
AUTHORS	Herbert, Kevin; Herlitz, Lauren; Woodman, Jenny; Powell, Claire; Morris, Stephen

VERSION 1 – REVIEW

REVIEWER	Karvandi, Elika University of Cambridge, Clinical Neurosciences
REVIEW RETURNED	10-Jan-2024

GENERAL COMMENTS	This is an excellent paper focusing on an important and timely issue. I have a few small comments, but overall, the paper is of good quality. When you write “health need”, it is worth considering whether using the term “healthcare need” is more appropriate. A more clear study objective would strengthen the paper. P 6 Line 49: “Despite some evidence of improvements, their effectiveness may have been impaired by inadequate compensation for additional deprivation-related health care needs via the core general practice funding formula.” This needs to be referenced. P 7 Line 26: is there a reason for the short timeframe for conducting the review (10 weeks)? Please could you explain a bit more about how this affected the papers included, analysis, and results. P 7 Line 53: why not qualitative studies? Could this have provided more insight? P 8 Line 11: “screened the first 100 hits in Google Scholar” better to say you screened the grey literature. The way it is currently phrased makes it seem too arbitrary. P9 Line 39-41: the number of studies based on location adds up to 23. Please could you clarify this (I presume one study covered two regions?). P9 line 54-56: it would be helpful to see this in a graph – maybe one showing the frequency of the age ranges covered.
--

	Could you add a couple of sentences discussing how the results in this review can be used. This would strengthen the paper and bring it together. Please review the references and ensure they are in accordance with the BMJ Open requirements (particularly the non-journal references).
--	--

REVIEWER	Kalaitzopoulou, Ioustini Aristotle University of Thessaloniki, Medical school
REVIEW RETURNED	05-Feb-2024

GENERAL COMMENTS	Well-structured work! There are a few comments to drop. Data sources: Google Scholar is not a good option for searching. You have already used four reliable databases. As it is already mentioned, you screened the first 100 hits, which eliminates the systematic searching, it is biased and also redundant. page 10/ lines 3-20: Are these lines necessary? It is known that you cannot apply this kind of data synthesis (e.i. meta-analysis) or have patient involvement. References: check for more recent articles to cite, if it's possible.
---

VERSION 1 – AUTHOR RESPONSE

1. Reviewer: 1

Ms. Elika Karvandi, University of Cambridge

Comments to the Author:

This is an excellent paper focusing on an important and timely issue. I have a few small comments, but overall, the paper is of good quality.

- a. When you write “health need”, it is worth considering whether using the term “healthcare need” is more appropriate.

Text checked and amended (throughout)

- b. A more clear study objective would strengthen the paper.

Objective has been revised (p3):

“To systematically map evidence to answer the research question: What is the relationship between the characteristics of children and young people (CYP) or their caregivers and primary care service use in the UK, taking into account underlying healthcare needs?”

- c. P 6 Line 49: “Despite some evidence of improvements, their effectiveness may have been impaired by inadequate compensation for additional deprivation-related health care needs via the core general practice funding formula.” This needs to be referenced.

Relevant reference has been added (p5)

- d. P 7 Line 26: is there a reason for the short timeframe for conducting the review (10 weeks)? Please could you explain a bit more about how this affected the papers included, analysis, and results.

The pragmatic approach to review design, how this affected the potential findings, and how the rigor of the review was maintained, is now addressed in the “Discussion” section (p16-17)

- e. P 7 Line 53: why not qualitative studies? Could this have provided more insight?

A scoping review of qualitative studies has been conducted in parallel, as part of the funded project under which this work falls.

Title: "Access to primary care for children and young people (CYP) in the UK: a scoping review of CYP's, caregivers', and healthcare professionals' views and experiences of facilitators and barriers". Submitted to BMJOpen (Manuscript ID bmjopen-2023-081620).

- f. P 8 Line 11: “screened the first 100 hits in Google Scholar” better to say you screened the grey literature. The way it is currently phrased makes it seem too arbitrary.

Text describing the literature search strategy has been amended (p7)

- g. P9 Line 39-41: the number of studies based on location adds up to 23. Please could you clarify this (I presume one study covered two regions?).

Layte & Nolan study covered primary care in both Scotland and Ireland. Text has been amended to clarify this (p8)

- h. P9 line 54-56: it would be helpful to see this in a graph – maybe one showing the frequency of the age ranges covered.

CYP age frequency distribution chart has been added (Figure 1, additional image file attached to submission) and referenced in “Study designs and samples” section (p8)

- i. Could you add a couple of sentences discussing how the results in this review can be used. This would strengthen the paper and bring it together.

Comments on this point have been added to Conclusion (p17)

- j. Please review the references and ensure they are in accordance with the BMJ Open requirements (particularly the non-journal references).

References style checked and updated where required.

2. Reviewer: 2

Dr. Ioustini Kalaitzopoulou, Aristotle University of Thessaloniki

Comments to the Author:

Well-structured work!

There are a few comments to drop.

- a. Data sources: Google Scholar is not a good option for searching. You have already used four reliable databases. As it is already mentioned, you screened the first 100 hits, which eliminates the systematic searching, it is biased and also redundant.

The authors understand that there is debate regarding the value of Google Scholar for systematic reviewing (e.g., <https://www.ncbi.nlm.nih.gov/pmc/articles/PMC4574933/>). They felt however that it was important to include a search of grey literature, due to the focus of the study, aspects of which (e.g., horizontal inequity) are of particular interest to organisations who produce research reports which would not fall within the scope of the key literature databases. Text referring to the grey literature search has been amended, and supporting references on the use of Google scholar in systematic literature searches has been added.

- b. page 10/ lines 3-20: Are these lines necessary? It is known that you cannot apply this kind of data synthesis (e.i. meta-analysis) or have patient involvement.

The authors decided to include the statement on data synthesis to relate the decision to provide a narrative synthesis of the evidence found.

Although patient involvement as not applicable in this case, the authors also wished to formally acknowledge all ethical considerations, as required by the project funder.

- c. References: check for more recent articles to cite, if it's possible.

An updated literature search will unfortunately not be possible, as the funding period for the project under which this work falls, has closed.

Reviewer: 1

Competing interests of Reviewer: None

Reviewer: 2

Competing interests of Reviewer: None

VERSION 2 – REVIEW

REVIEWER	Karvandi, Erika University of Cambridge, Clinical Neurosciences
REVIEW RETURNED	16-Apr-2024
GENERAL COMMENTS	The manuscript has been improved - well done on getting this done!